# Hypoxia Dictates Metabolic Rewiring of Tumors: Implications for Chemoresistance

**DOI:** 10.3390/cells9122598

**Published:** 2020-12-04

**Authors:** Dimas Carolina Belisario, Joanna Kopecka, Martina Pasino, Muhlis Akman, Enrico De Smaele, Massimo Donadelli, Chiara Riganti

**Affiliations:** 1Department of Oncology, University of Torino, via Santena 5/bis, 10126 Torino, Italy; dimascarolina.belisario@unito.it (D.C.B.); joanna.kopecka@unito.it (J.K.); martina.pasino530@edu.unito.it (M.P.); muhlis.akman@unito.it (M.A.); 2Department of Experimental Medicine, Sapienza University of Roma, 00185 Roma, Italy; enrico.desmaele@uniroma1.it; 3Department of Neurosciences, Biomedicine and Movement Sciences, Section of Biochemistry, University of Verona, 37134 Verona, Italy; massimo.donadelli@univr.it

**Keywords:** hypoxia, cancer, metabolic reprogramming, chemoresistance

## Abstract

Hypoxia is a condition commonly observed in the core of solid tumors. The hypoxia-inducible factors (HIF) act as hypoxia sensors that orchestrate a coordinated response increasing the pro-survival and pro-invasive phenotype of cancer cells, and determine a broad metabolic rewiring. These events favor tumor progression and chemoresistance. The increase in glucose and amino acid uptake, glycolytic flux, and lactate production; the alterations in glutamine metabolism, tricarboxylic acid cycle, and oxidative phosphorylation; the high levels of mitochondrial reactive oxygen species; the modulation of both fatty acid synthesis and oxidation are hallmarks of the metabolic rewiring induced by hypoxia. This review discusses how metabolic-dependent factors (e.g., increased acidification of tumor microenvironment coupled with intracellular alkalinization, and reduced mitochondrial metabolism), and metabolic-independent factors (e.g., increased expression of drug efflux transporters, stemness maintenance, and epithelial-mesenchymal transition) cooperate in determining chemoresistance in hypoxia. Specific metabolic modifiers, however, can reverse the metabolic phenotype of hypoxic tumor areas that are more chemoresistant into the phenotype typical of chemosensitive cells. We propose these metabolic modifiers, able to reverse the hypoxia-induced metabolic rewiring, as potential chemosensitizer agents against hypoxic and refractory tumor cells.

## 1. Introduction

Depending on the tissue type, there is a wide variability in the oxygen (O_2_) levels, ranging from 9.5% (72.0 mmHg) in kidneys [1], 7.6–6.8% (57.6 mmHg-51.6 mmHg) in gastrointestinal tract [2,3], 5.6% (42.8 mmHg) in lungs [4], 5.4% (40.6 mmHg) in liver [5], and 4.4% (33.8 mmHg) in the brain [6]. O_2_ levels below these values are considered hypoxic. Physiological hypoxia implies an adaptive and homeostatic response, such as vasodilation and/or up-regulation of hypoxia response genes, to maintain stable levels of O_2_. On the contrary, in pathological hypoxia, the homeostatic mechanisms do not compensate adequately the falling in O_2_ levels [7].

The fast rate of growth in solid tumors makes them susceptible to O_2_ shortage in poorly vascularized areas and leads to the development of intratumoral hypoxic regions [8,9]. Neo-angiogenesis is a compensative response to intratumoral hypoxia. However, the tumor vasculature is composed of leaky vessels with chaotic architecture and easy tendency to collapse under the pressure of growing tumor and stromal cells [10]. Although the new vessels formed supply O_2_, the irregular architecture and the vascular collapse reduce the oxygenation in many tumor areas that reach 1–1.3% (8–10 mmHg) O_2_ pressure [7,11,12]. The cycling between vessels formation and collapse induces fluctuation of O_2_ levels, producing repeated cycles of hypoxia and normoxia within specific areas of tumor bulk [13]. Moreover, the absence of lymphatic drainage induces intermittent vascular collapse and creates, temporarily and acutely, hypoxic areas that have been proposed to contribute to progression and/or relapse [14]. Chemotherapy used in cancer treatment can further damage blood vessels, contributing to generate areas with chronic hypoxia in the tumor mass [11]. Microregions with very low (i.e., near to zero) levels of O_2_ are heterogeneously distributed within the tumor bulk, with a prevalence of better oxygenated areas, characterized by a high rate of cell division and tumor growth around the capillaries. The newly generated cells often migrate towards the regions far from vessels [15]. Indeed, hypoxia increases the invasive potential of cells by affecting the extracellular matrix (ECM) [16,17,18], e.g., by stimulating the paracrine secretion of soluble factors that generate a fibrotic and stiff ECM, favorable to cell spreading [19,20]. Notably, even when re-exposed to O_2_, hypoxic tumors maintain high the expression of hypoxia-sensitive genes inducing metastasis and resistance to oxidative stress [21], conserving a “hypoxic memory” that determines a peculiar aggressiveness [22].

Hypoxia not only affects neoplastic cells, but also implies changes in metabolism and functions of infiltrating cells, such as cancer-associated fibroblasts (CAFs) and tumor-associated macrophages (TAMs). These changes may impair or favor the neoplastic growth, producing cellular quiescence, differentiation, apoptosis, or necrosis, depending on the degree, persistence, and severity of hypoxia. The tolerance to hypoxia, i.e., the ability to enter a quiescent but viable status, determines the persistence of hypoxia-tolerant cells that are aggressive and hard to be eradicated pharmacologically [23].

As a consequence of the different oxygenation, solid tumors are metabolically heterogeneous: better oxygenated regions rely on mitochondrial oxidative phosphorylation (OXPHOS), while hypoxic areas are more dependent on anaerobic metabolism [21]. This metabolic reprogramming is coordinated by the hypoxia-inducible factors (HIF) family. According to our present knowledge, up to 2% of the human genome is modulated by HIF transcription factors [24]. This review will focus on the metabolic rewiring induced by hypoxia, on the implications of such rewiring in tumor progression and chemoresistance, on the new therapeutic opportunities that may emerge with a deep knowledge of the metabolic reprogramming occurring in hypoxia.

## 2. Hypoxia-Inducible Transcription Factors and Hypoxia-Targeted Genes

HIF is a heterodimer formed by two subunits, the O_2_-regulated HIF-α subunit, and the O_2_-independent, constitutively expressed aryl hydrocarbon receptor nuclear translocator (ARNT), also called HIF-β [24,25,26,27,28]. Three HIF-α homologues have been identified: HIF-1α, HIF-2α, and HIF-3α. In humans, HIF-1α is expressed in most tissues, HIF-2α is mainly expressed in kidneys, brain, lungs, liver, gastrointestinal tract, pancreas and heart [29]. HIF-3α is expressed in several tissues [30], but has a different protein structure [31,32].

The regulation of stability and activity of HIF-1α and HIF-2α is dependent on O_2_ levels [33]. Indeed, the half-life of the HIF-α subunits in the presence of 21% O_2_ is less than 5 min, but it increases to 60 min as the O_2_ concentration decreases to 1%. Under normoxia, HIF-α subunit is hydroxylated by prolyl hydroxylase domain (PHD)-containing proteins [33]. This is the result of the interaction between HIF-α and the von Hippel Lindau tumor suppressor protein (pVHL), a part of E3-ubiquitin ligase complex, which recognizes two hydroxylated proline residues of HIF-α (P402 and P564): upon hydroxylation, HIF-α is primed for polyubiquitination and proteasomal degradation [34,35]. Furthermore, HIF-α subunits are subjected to asparagine hydroxylation at N803 residue: this modification hinders the binding of HIF-1α with the general co-activator p300/CREB binding protein (CBP)complex [36]. By limiting O_2_ availability, hypoxia inhibits all of the hydroxylation processes, restoring its transcriptional efficiency [37,38].

Moreover, microRNAs (miRNAs) play an important role in HIF-1α stabilization and/or in its down-stream effects [39]. Indeed, specific miRNAs, termed hypoxia-regulated miRNAs (HRMs), are differentially expressed in response to hypoxia [40], and modulate angiogenesis, apoptosis, proliferation, metastasis, and chemoresistance [41].

Besides hypoxia and miRNAs, the stability and activation of HIF-1α and HIF-2α in tumors is also favored by growth factors, cytokine, and oxidative stress [42]. Several receptor tyrosine kinases (RTKs), such as epidermal growth factor receptor (EGFR) family members (erbB1 and erbB2/HER2/Neu), insulin-like growth factor-1 receptor (IGF-1R), stem cell factor (SCF)/KIT receptor, Notch, interleukin-6 receptor (IL-6R) and transforming growth factor-β receptor (TGF-βR) activate HIF-1α. The cooperation between their downstream transducers, such as phosphatidylinositol 3′-kinase (PI3K)/Akt/molecular target of rapamycin (mTOR), Ras/mitogen-activated kinase (MEK)/extracellular signal-regulated kinase (ERK), nuclear factor-κB (NF-κB), signal transducer and activator of transcription 3 (STAT3) axes also increase HIF-1α, by inducing a phosphorylation on serine residues that make the protein less susceptible to hydroxylation and ubiquitination [9,43]. Additionally, the inactivation of pVHL [37], phosphatase and tensin homolog deleted on chromosome 10 (PTEN) [44] and p53 [45]—three onco-suppressors often inactivated or deleted in cancer cells—may block HIF degradation and/or further enhance PI3K/Akt activation.

The main target genes of HIF-1α are erythropoietin (EPO), the pro-angiogenic vascular endothelial growth factor (VEGF), the pro-invasive factors metalloproteinase 9 (MMP9) and urokinase-type plasminogen activator (uPA) [46,47], the glucose-metabolizing enzymes—glucose transporters 1 and 3 (GLUT1 and GLUT3), glycolytic enzymes, such as hexokinase (HK), phosphofructokinase-1 (PFK1), aldolase, triose-phosphate isomerase (TPI), glyceraldehyde 3-phosphate dehydrogenase (GAPDH), enolase, lactate dehydrogenase A (LDHA), pyruvate kinase M2 (PKM2), and pyruvate dehydrogenase kinase 1 (PDK1) [45,48], the amino acids (AA) transporters xCT (SLC7A11), and L-type amino acid transporter 1 (LAT1/SLC7A5) [49,50], the chemoresistance inducer multidrug resistance 1 (MDR1), which encodes for the drug efflux transporter P-glycoprotein (Pgp) [51]. Particularly, GLUT1 is an essential element for glucose uptake in cancer cells and it is a marker of more aggressive tumors [48]. Besides glucose, tumor also needs AA and lipids [52] to meet the increased demand of building blocks despite the poorly organized intratumoral vasculature [53], and hypoxia affects both these metabolisms.

The majority of the transcriptional programs activated by hypoxia in tumors are due to the activity of HIF-1α and HIF-2α. The function of HIF-3α remains to be fully elucidated, although it is known to suppress HIF-1α- and HIF-2α-mediated gene expression and to upregulate a different set of genes [30], thus, finely tuning the hypoxic response mediated by HIF-1α- and HIF-2α [54].

## 3. Hypoxia Affects Metabolic Rewiring in Solid Tumors

One of the hallmarks driving aggressiveness in cancer is the reprogramming of energy metabolism [55]. Glycolysis becomes the predominant energetic pathway, in the presence or absence of O_2_, ensuring survival and proliferation [56]. This feature is known as the “Warburg effect” since Otto Warburg observed for the first time that cancer cells have increased conversion of glucose to lactate compared to normal cells [57]. In parallel to the enhanced glycolysis, cancer cells are equipped with several membrane proteins—such as the monocarboxylate transporters (MCTs) and carbonic anhydrases (CAs)—buffering the end products of anaerobic glycolysis, such as lactate and H^+^. These metabolic changes are initiated by oncogenes and perpetuated by the homeostatic alterations of the tumor microenvironment (TME) [58,59]. Hypoxia is a key factor promoting such metabolic rewiring, affecting different pathways and compartments (Figure 1).

### 3.1. Glucose Transport and Glycolysis

Cells of solid tumors are often glucose-addicted, although the glycolytic pathway followed by lactic fermentation has a lower efficiency in terms of glucose/ATP ratio, compared to the glycolysis followed by tricarboxylic acid (TCA) cycle and oxidative phosphorylation (OXPHOS). Indeed, although cancer cells take up glucose at higher rates than normal tissues, they oxidize less glucose in TCA and OXPHOS, regardless of the presence of O_2_ [8]. Despite the lower energetic yield, glycolysis is important in cancer cells because it is a crucial source of intermediates used for the synthesis of macromolecules, required for proliferation and migration [60]. Indeed, there is a strong linkage between increased glucose uptake and tumor aggressiveness [61].

Na^+^-coupled glucose transporters (SGLTs) contribute to glucose uptake in non-transformed cells, but they are not modified by hypoxia. The main transporters involved in glucose uptake in cancer cells belong to the GLUTs/SLC2A family [61]. An increased expression of the high affinity GLUT1 protein has been correlated to metastases and poor prognosis [53,61], appearing a hallmark of aggressive tumors. Both hypoxia-dependent and independent mechanisms regulate GLUT1 expression. GLUT1 and GLUT3 are direct target of HIF-1α [56,62], providing a link between hypoxia, Warburg effects, tumor growth and progression. Moreover, activated Ras and c-Myc, as well as the pro-tumorigenic antigens E6 and E7 of human papilloma virus (HPV), increase GLUT1 expression [63], determining a hypoxia-independent increase of glucose uptake and metabolism during the early stages of tumor progression.

Besides glucose uptake, HIF-1α accelerates the transformation of glucose into lactate, being a transcriptional inducer of the glycolytic enzymes [9]. In parallel, it prevents the pyruvate mitochondrial metabolism, by upregulating PDK1, which phosphorylates and inactivates the catalytic subunit of pyruvate dehydrogenase (PDH) [64]. This event reduces the oxidation of pyruvate to Acetyl Coenzyme A (AcCoA) and the subsequent entry into the TCA cycle [64]. This is part of an adaptive strategy to the hypoxic environment, which leads cancer cells to limit the TCA fueling by AcCoA and the subsequent OXPHOS in the absence of adequate O_2_ levels.

### 3.2. Lactate Metabolism and Acidosis

As a result of the increased anaerobic glycolysis, hypoxic areas of tumors produce huge amounts of lactic acid, which easily dissociates into lactate and H^+^, promoting the acidosis of TME [65]. Originally considered as a waste product indicative of stress, lactate has been re-evaluated as an energetic substrate for cancer cells, as well as a signal molecule promoting tumor progression [65,66]. Lactate is exported by proteins of MCT family, although their expression is rather variable within tumor bulk and depends on the different hypoxia degree and cell metabolism. MCT1 transports lactate in well-oxygenated areas but not in hypoxic regions, whereas MCT4 is the prevalent isoform. In hypoxic areas, MCT4 is co-expressed with HIF-1α [67,68], likely because it is a target gene of HIF-1α [69]. By contrast, the linkage between MCT1 expression and HIF-1α activity is controversial and tumor-dependent [70,71]. This pattern of distribution of MCTs determines a peculiar metabolic symbiosis within different areas of tumors, or between cancer cells and non-transformed cells, such as CAFs. Indeed, hypoxic cells, mainly localized in the tumor core, export lactate via MCT4. Lactate, taken up via MCT1 by more oxygenated cells—prevalently localized in tumor periphery—is transformed into pyruvate, fueling the oxidative metabolism of these cells [67,72] or promoting biosynthetic processes [65,73]. Indeed, lactate is imported within mitochondria via specific MCTs; here, a mitochondrial LDH isoform transforms lactate into pyruvate, which can act both as an energetic substrate or as a building block [65,73]. The importance of this metabolic symbiosis between different populations of cancer cells is demonstrated by the dramatic necrosis occurring in the hypoxic core of solid tumors treated with MCT1 inhibitors [67] that prevent the lactate shuttle from hypoxic to normoxic areas. A similar symbiosis occurs between cancer cells and CAFs, because also stromal cells in hypoxic areas activate the same HIF-1α-driven programs of cancer cells, i.e., they up-regulate GLUT1, glycolytic enzymes, and MCT4 [74]. The lactate exported by hypoxic CAFs via MCT4 is readily taken up by cancer cells via MCT1 and fuel the TCA/OXPHOS flux in tumor areas with a sufficient supply of O_2_, producing the so-called “reverse Warburg effect” [74,75]. Notably, the increased transformation of lactate into pyruvate also increases the NADH/NAD^+^ ratio in cancer cells. This event promotes the sirtuin 1 (SIRT1)-induced deacetylation and activation of the peroxisome proliferator-activated receptor gamma coactivator-1α (PGC-1α), an inducer of mitochondrial biogenesis [76]. As a result, the lactate produced by hypoxic CAFs equips cancer cells with multiple tools, i.e., energetic substrates, NADH, and mitochondrial mass, which allow the switch towards an oxidative metabolism if O_2_ is adequate. Since tumors undergo multiple cycles of hypoxia and re-oxygenation [13], this switch may help cancer cells to rapidly adapt to the changes in O_2_ levels, continuing their growth notwithstanding mutable environmental conditions.

Besides its role as energetic substrate, lactate also promotes tumor progression and invasion. For instance, it stimulates the synthesis of VEGF by endothelial cells, promoting neo-angiogenesis [65,77]. By increasing the secretion of TGF-β, lactate favors the degradation of extracellular matrix by MMPs, promoting cancer cell invasion [78] and contributing to tumor progression.

The co-secretion of H^+^ with lactate by MCT symporters induces a strong acidification of TME, which reaches values between 6 and 6.5. Such acidosis determines metabolic reprogramming [79] and invasiveness [80,81] as well.

Since the glycolytic pacemaker enzyme PFK1 is inhibited by low pH, it is not surprising that cells growing in acidosis reduce the glycolytic flux, increasing the glucose metabolism in the pentose phosphate pathway (PPP) [80]. This effect is obtained also by the transcriptional increase of glucose 6-phosphate dehydrogenase (G6PD), the pacemaker enzyme of PPP [79], and by the down-regulation of several HIF-1α-target glycolytic genes [82] during acidosis. Similarly, acidosis up-regulates glutaminase and, consequently, increases glutaminolysis. The oxidation of glutamine into glutamate and α-ketoglutarate (αKG) fuels the TCA cycle, generating a mitochondrial-dependent production of ATP [79]. This is relevant to determine cell survival under mild hypoxic conditions. Indeed, acidosis reshapes mitochondria cristae and maximizes the efficiency of ATP production [83], even during hypoxia.

An increased fatty acids (FAs) oxidation (FAO) is also observed in acidosis [82]. Such increase has at least two consequences: on the one hand it fuels the TCA cycle; on the other hand the high levels of AcCoA promotes an inhibitory acetylation of the complex I of electron transport chain (ETC), reducing the production of reactive oxygen species (ROS) by OXPHOS [82]. This mechanism, together with the increased production of NADPH via PPP [80], prevents the oxidative damages of cells growing in a hypoxic environment. Moreover, the abundance of AcCoA induces the acetylation of SIRT 1/6. By inducing histone deacetylation, SIRT1/6 downregulate AcCoA carboxylase 2 (ACC2), removing the negative regulation exerted by malonyl CoA on mitochondrial FAO [82]. SIRT1 also activates HIF-2, which in turn upregulates enzymes involved in both oxidative and reductive metabolism of glutamine [84]. The reductive carboxylation of glutamine (see paragraph 3.3. for details) generates citrate, an excellent source for FA synthesis. Moreover, acidosis increases the transcriptional activity of the lipogenic factor sterol regulatory element-binding protein 2 (SREBP2), which promotes FA and cholesterol synthesis, favoring cell proliferation and tumor progression [85]. The co-existence of FAO and FA synthesis, oxidative and reductive metabolism of glutamine in acidosis, suggests that acidic cancer cells rely on a metabolic balance, paradoxically based on futile cycles. This situation is likely consequent to the progressive adaptation to the hypoxic and acidic environment that triggers multiple metabolic pathways to spare energy and synthesize building blocks. Disrupting such cycles, by using FAO or glutaminolysis inhibitors [82], may be a promising anticancer strategy against tumor acidosis.

Besides inducing a strong metabolic reprogramming, hypoxia and lactic acidosis favor the self-maintenance of cancer stem cells (CSCs) [86,87], and activate the pro-metastatic epithelial-mesenchymal transition (EMT) program. These two events are deeply interconnected: indeed, EMT increases the migratory and invasive properties of cancer cells, as well as the acquisition of stem cell-like features [88,89]. An acidic environment stimulates the autocrine production of TGF-β2. First, the TGF-β2-dependent signaling induces the phosphorylation and acetylation of SMAD family member 2 (Smad2), a driver of EMT [81]. Second, TGF-β2 activates the protein kinase C ζ (PKCζ), which increases the FA uptake via the scavenger receptor CD36 and the accumulation as triglycerides (TGs) in lipid droplets [81]. These findings are in keeping with the increased lipogenesis observed in acidosis [83]. Anoikis resistance and invasion are prevented not only by inhibitors of Smad2, but also by inhibitors of lipogenesis [81]. This observation highlights an unexpected bridge between metabolic reprogramming induced by acidosis and tumor progression.

Finally, hypoxia and lactic acidosis alter the TME, promoting an immune-suppressive environment [75]. Hypoxia increases the polarization of TAMs towards a tumor-permissive M2 phenotype and the amount of myeloid-derived suppressive cells (MDSC) infiltrating the tumor [77]. Lactate produced in hypoxic TME has been implicated in the impaired differentiation of monocytes into dendritic cells (DCs) [90], in the loss of the cytolytic functions by natural killer (NK) cells [91], in the increases of M2 macrophages [92] and MDSC that inhibit NK activity [93]. Not only lactate produced by tumor cells, but also lactate originated by the immune-infiltrating cells contributes to this immune-suppressive environment. For instance, anti-tumor M1-polarized TAMs are actively glycolytic in hypoxic areas [93], but MCT4 inhibitors that block the export of lactate from M1 TAMs induce the polarization towards a M2 phenotype [90]. Active DCs are also highly glycolytic and export huge amounts of lactate [93]. As observed for TAMs, blocking the export of lactate from DCs induces the acquisition of an immune-suppressive phenotype, as demonstrated by the lower ability of recruiting CD8^+^ T-lymphocytes [90]. Activated T-cells highly rely on anaerobic glycolysis, as suggested by the upregulation of GLUT1 during T-lymphocyte activation [94]. This process, which is enhanced by hypoxia, determines the selection of CD8^+^ T-lymphocytes clones with strong anti-tumor activities, characterized by high expansion and production of interferon-γ (IFN-γ) [95]. However, an excessive production of lactate by tumor cells inhibits the export of lactate by MCT1 present on CD8^+^ T-lymphocytes: the intracellular acidosis of lymphocytes suppresses the cytotoxic potential and the production of anti-tumor cytokines [96]. As already described for lactate, the high production of H^+^ from cancer cells may affect the efficiency of immune cells in hypoxic areas. For instance, highly glycolytic tumors producing H^+^ impair the activation of NK and CD8^+^ T-lymphocytes because the low pH inhibits the transcriptional activity of nuclear factor of activated T-cells (NFAT) [97], a master regulator of T-cells development and activities. Moreover, cancer cells and CD8^+^ T-lymphocytes compete for glucose: in hypoxic and highly glycolytic tumors, the scarcity of glucose left to lymphocytes impair their proliferation and production of IFN-γ [98]. Since the immune-checkpoint PD-1L increases the glycolysis in cancer cells by activating Akt/mTOR pathway, the use of immune-checkpoint inhibitors may relieve the competition for glucose between cancer cells and T-lymphocytes [98], restoring the anti-tumor efficiency of the latter.

Overall the higher export of lactate and the consequent acidosis characterizing hypoxic tumor areas, the metabolic cross-talks between cancer cells and non-transformed cells within TME (e.g., CAFs and immune cells) favor proliferation, invasiveness and immune-escape, reducing the efficacy of anti-tumor pharmacological treatments.

### 3.3. Amino Acid Transport and Glutaminolysis

As already mentioned, HIF-1α and HIF-2α upregulate the AA transporters xCT (SLC7A11) and LAT1 (SLC7A5) [49,50]. While the former is a Na^+^-independent exchanger of cystine and glutamate [99], the latter is responsible for the transport of large neutral and essential AAs [100]. Both AA transporters form a heterodimer with the CD98 glycoprotein, which acts as a chaperone promoting stabilization, trafficking, and insertion of these transporters into the plasma membrane [99].

By importing cystine necessary for the biosynthesis of glutathione (GSH), the main intracellular anti-oxidant metabolite, xCT plays a key defensive role against oxidative stress and chemotherapy [101]. Although some tissues (e.g., the liver, kidney, and pancreas) can synthesize cysteine from homocysteine and serine by a transsulfuration pathway [102], in most cells an adequate supply of cysteine is granted only by the cystine import via xCT. Extracellular cysteine is unstable and quickly oxidized to cystine, the substrate of xCT: the higher reducing intracellular conditions favor the conversion of cystine into cysteine and its incorporation into GSH. Notably, the HIF-1α-induced increase in xCT expression and GSH synthesis favors the survival of the CSC population in triple negative breast cancer (TNBC) [49], suggesting that during tumor initiation and/or recurrence the HIF-1/xCT/GSH axis has a critical role in the protection against oxidative stress and in the expansion of stem cell subclones.

Several tumor types also overexpress LAT1 [50,103], which is upregulated by HIF-2α [50]. Concurrently HIF-2α activates mTOR, a sensor of the cellular energy status and of the AA availability, and a stimulator of protein translation and cell proliferation [104]. mTOR interacts with other proteins to form two distinct complexes, termed mTOR complex 1 (mTORC1) and mTOR complex 2 (mTORC2), the former of which is activated by LAT1 [105]. Hence, HIF-2α/LAT1 axis plays a dual role, by ensuring the uptake of essential AA within cancer cells, notwithstanding the nutrient-depleted TME, and by favoring their prompt incorporation into proteins necessary for survival and proliferation.

Another important reprogramming of cellular energy metabolism in solid tumors is related to the increased glutaminolysis [106]. Glutamine is an abundant AA, which is involved in energy production by fueling TCA cycle, in anabolic pathways as source of building blocks for nucleotides, proteins and plasma membrane glycolipids/glycoproteins, and in oxide-reductive homeostasis, as precursor of GSH [107]. Besides, glutamine favors lipogenesis by at least two mechanisms. First, αKG metabolized in the TCA cycle generates malate that is transformed into pyruvate by the malic enzyme: this step generates reducing equivalents (NADPH), necessary for the synthesis of FAs and sterols [108]. Second, αKG can be carboxylated to isocitrate by isocitrate dehydrogenase (IDH) [109]. IDH1 is the cytosolic isoform reversibly transforming αKG into isocitrate and contributing to the reductive metabolism of glutamine; IDH2 mainly transforms isocitrate into αKG in mitochondria [110], contributing to the oxidative metabolism of glutamine via TCA cycle. Hypoxia promotes the activity of IDH1, which generates isocitrate and NADPH, while it reduces the oxidative catabolism of glutamine via IDH2 [111]. Isocitrate is then transformed into citrate and AcCoA via cytosolic aconitase and citrate lyase, providing carbon source for lipid synthesis [111]. NADPH supports the reductive steps in FA and cholesterol synthesis, as well as the maintenance of oxide-reductive balance [112]. The prevalence of reductive glutamine metabolism in hypoxia has been explained by the inhibition of HIF-1α on PDH [64,111], which reduces the TCA cycle flux and consequently limits the oxidative metabolism of glutamine via IDH2. Such shift in glutamine metabolism makes the AcCoA produced by reductive glutamine carboxylation the most important source for de novo lipogenesis in cells grown under hypoxia or with high levels of HIF-1α. As proof of concept, pVHL-deficient renal cancer cells preferentially utilize glutamine metabolism for lipid synthesis even at normal O_2_ levels [111,113]. Although important, glutamine is not the only AA supporting lipid synthesis in hypoxia. For instance, the serine catabolism via serine hydroxymethyl transferase 2 (SHMT2) also provides NADPH [114]. Indeed, HIF-2 upregulates SHMT2 that transforms serine into glycine and produces at the same time methylene-tetrahydrofolate (THF). The oxidation of the latter to 10-formyl-THF by methylene-THF dehydrogenase produces NADPH [115], which neutralizes ROS and supports lipid synthesis in hypoxic tumors.

### 3.4. Fatty Acid Metabolism

Protein and DNA synthesis, commonly accelerated in growing tumors, are energy-consuming processes [116]. One of the main energy source is represented by endogenous lipids. Moreover, lipids are also necessary building blocks in the synthesis of plasma membrane and intracellular membranes of rapidly dividing cells. Interestingly, HIF-1α up-regulates both fatty acid synthase (FASN), the multimeric complex deputed to FA synthesis, and lipin 1 (LPIN1), a phosphatidate phosphatase that transforms phosphatidic acid into diacylglycerol, i.e., catalyzing the penultimate step of TG synthesis [117,118]. This is an important adaptive response of tumors to hypoxia, since it facilitates FA and TG storage [119], meeting both the energy and synthetic requirements of proliferating cells. The active endogenous synthesis of lipids also explains the common observation of intracellular lipid droplets, particularly abundant in hypoxic tumors [120]. The de novo synthesis of FAs supplies more than 93% of the FAs containing in the TG and phospholipids (PLs) of tumors [121]. The low consumption of AcCoA within TCA caused by the upregulation of PDK1 and the reductive carboxylation of glutamine into citrate favored by hypoxia, are additional factors promoting the cytosolic supply of AcCoA for FASN.

On the other hand, AcCoA can be also generated by FAO. Since cancer cells have increased lipogenesis in hypoxia, one should expect a reduced lipid catabolism to sustain rapid cell proliferation. Nevertheless, the scenario is variegate and sometimes contradictory. Indeed, it is a common observation that the acidosis induced by hypoxia up-regulates both FA synthesis and FAO [82]. In some tumors, the activation of FAO negatively affects growth and progression [121], in other tumors, FAO is down-regulated, and its increase reduces cell proliferation [122]. By contrasts, an increased rate of FAO has been associated with an increased survival of cancer cells in stressful conditions [123,124], although the mechanisms remain unclear. Other authors have proposed that FAO is the preferential source of energy in quiescent cancer cells that have a finely tuned balance between FA synthesis and FAO, and are less dependent on glucose and glutamine [125]. Moreover, the type of FAs oxidized impacts on the cell fate. Indeed, cancer growth is suppressed by the expression of long-chain acyl-CoA dehydrogenase (LCAD), but not by the expression of the medium-chain acyl-CoA dehydrogenase (MCAD). While both LCAD and MCAD are essential for the first step of FAO in mitochondria, their substrates and products are different. LCAD preferentially oxidizes long unsaturated FAs; MCAD preferentially oxidizes saturated FAs. Hence, the suppression of LCAD increases the accumulation of unsaturated FAs: this is a mechanism inhibiting the PTEN oncosuppressor by upregulating miR-21 in hepatocellular carcinoma [126], but it has not been reported in other tumor types.

HIF-1α decreases the expression of carnitine palmitoyltransferase 1A (CPT1A), the rate limiting-enzyme of FAO [127]. CPT1A mRNA levels are lower in slowly-proliferating HER2-positive and luminal B breast cancer subtypes than in highly proliferating and metastatic TNBC [128], suggesting that in breast cancer FAO is active more in indolent and slow progressing subtypes. HIF-1α also down-regulates acyl-CoA dehydrogenases (ACAD), but in this case the reduction of FAO flux is not paralleled by reduced cell survival and proliferation [129], raising doubts on the biological impact of such inhibition. Overall, while the role of HIF-1α in rewiring glucose and AA metabolism is clearly directed to support tumor growth, the impact of hypoxia on lipid metabolism and its biological meaning is not univocal. The analysis of the differences in tumor types, in enzymatic sets involved in lipid synthesis and oxidation, in driving oncogenes or oncosuppressor genes, in hypoxia degree may explain the contrasting evidences and may help to predict if the effects of HIF-1α on lipid anabolic or catabolic pathways produce cell death or survival.

### 3.5. Mitochondria in Hypoxic Tumors

Over the last three decades, an attenuated mitochondria metabolism associated to hypoxia has been reported in several pathological conditions, including cancer [114].

Since hypoxia triggers several mechanisms to adapt tumor cells to low O_2_ levels and mitochondria are the major O_2_ consumers, the physiological consequence of hypoxia is a reduced mitochondrial OXPHOS. Indeed, in solid tumors hypoxia reduces the activity of the TCA cycle by up-regulating PDK1 [64,124], inhibits the expression of ETC components [130], represses mitobiogenesis by increasing the Hes-related family BHLH Transcription Factor With YRPW Motif (HEY) protein [131], alters the fusion/fission balance [132], increases mitochondrial autophagy by activating the pro-mitophagic factor BCL2 Interacting Protein 3 (BNIP3) gene [133]. The sum of these events is a quantitative reduction of mitochondria number and a reduction in their energetic efficiency. On the other hand, the metabolic symbiosis between hypoxic CAFs or tumor cells and well oxygenated cells [65,73,74], supported by the lactate shuttle, favors the OXPHOS metabolism in tumor areas, with sufficient levels of O_2_ or during re-oxygenation phases. Lactate produced by CAFs increases the mitochondrial mass by upregulating PGC-1α [76], and mild acidosis associated with hypoxia reshapes mitochondria favoring the optimal conditions for OXPHOS and ATP synthesis [83]. Consequently, tumor areas characterized by mild hypoxia or acidosis basically rely on an anaerobic metabolism, but are well equipped to switch towards a mitochondrial energetic metabolism, adapting to the continuous and mutable changes of O_2_ levels.

The scarcity of O_2_ also leads to an incomplete reduction at complex IV of ETC, increasing the production of partially reduced O_2_-derived species, i.e., ROS. Mitochondrial ROS (mtROS) can increase the rate of mutations in mitochondrial DNA (mtDNA) that encodes for 13 subunits of complexes I to V [134]. The presence of mutated mtDNA, along with the reduced content of mtDNA frequently observed in tumors with a defective mitobiogenesis [135], are other reasons accounting for the reduced OXPHOS in hypoxic tumors. This phenotype has been associated with increased invasiveness and metastases, and with a poor clinical outcome across several cancer types [135]. The presence of surrounding stroma can partially prevent the loss of mtDNA, as demonstrated by the injection of mtDNA-deprived tumor cells in mice, which are progressively enriched in mtDNA provided by the murine stroma. This process is associated with the recovery of mitochondrial respiratory functions and to the assembly of new mitochondrial structures [17]. This study suggests that the crosstalk within different cell populations of the TME may counteract the attenuation of mitochondrial functions induced by hypoxia. As mentioned before, TME is characterized by repeated cycles of transient hypoxia followed by re-oxygenation, a condition that—like the ischemia-reperfusion condition—increases the generation of ROS and the ROS-dependent cell death [64]. Given the harmful role of ROS, the metabolic reprogramming that limits the level of AcCoA and the TCA cycle flux in hypoxia could be regarded as an adaptive strategy to limit the production of endogenous and dangerous ROS. Although the reduction of ROS via the reduction of TCA flux and FAO is considered protective by some studies [127,136], ROS inactivate the PHD proteins, thereby increasing HIF-1α [137,138]. Experiments of gene knock-down have highlighted that the main sources of mtROS stabilizing HIF-1α are complex II [139], complex III [140], complex I and TCA cycle enzymes [141]. In addition, specific oncogenes, such as Ras and c-Myc have been demonstrated to increase O_2_ uptake and ETC: if these oncogenes are active in hypoxic tumors, the increased ETC coupled with an uncompleted O_2_ reduction increase mtROS [142]. Instead, if ROS are maintained at sub-cytotoxic levels by buffering anti-oxidant enzymes, ROS contribute to HIF-1α stabilization [42]. Hence, if ROS levels are below the cytotoxic threshold to induce cell death, they become advantageous molecules for cancer cells, by increasing the amount of HIF-1α and stimulating the metabolic adaptation to hypoxia, as well as the pro-survival and pro-invasive programs induced by hypoxia.

## 4. Hypoxia Induces Chemoresistance by Pleiotropic Mechanisms

Within the same tumor, there are normoxic and hypoxic areas: such heterogeneity determines metabolic niches that are associated with more chemosensitive or more chemoresistant cells. The latter are usually characterized by higher stemness and higher tendency to relapse [143]. There are no fixed division between normoxic and hypoxic areas, but rather a continuum of “metabolically-modulated niches”, where cells metabolically adapted to survive to changing environmental conditions become more aggressive and resistant [144]. By activating such adaptive responses, cancer cells are facilitated not only to survive during O_2_ oscillations, but also pushed towards progression, resistance to chemotherapy and radiotherapy [35]. Several factors related or unrelated to the metabolic rewiring induced by hypoxia are at the basis of chemoresistance.

Among the metabolism-independent factors (Figure 2), one simple reason of the poor efficacy of chemotherapy is the tumor architecture. Indeed, in solid tumors, the cells in hypoxic regions, usually far from the vessels, have limited delivery of chemotherapeutic agents because of the abnormal vascularization [145,146]. Moreover, HIF-1α upregulates multiple genes that contribute to chemoresistance. One of this is *MDR1*, encoding for Pgp [51]. Pgp belongs to the adenosine triphosphate (ATP)-binding cassette (ABC) transporters family and effluxes several chemotherapeutic drugs unrelated for structure and mechanism of action, contributing to a multidrug resistance (MDR) phenotype [147]. The HIF-1α-dependent up-regulation of Pgp has been documented in liver [148], larynx [149], lung [150], and breast [151,152,153] cancers, as well as in malignant pleural mesothelioma [154] and B-cells chronic lymphocytic leukemia [155]. The broad range of tumors showing this behavior suggests that this is a quite conserved mechanism. Moreover, other ABC transporters involved in drug efflux, such as MDR related protein 1 (MRP1) [156], lung resistance protein (LRP) [157] and breast cancer resistance protein (BCRP) [158,159], are under the transcriptional control of HIF-1α and HIF-2α, enlarging the number of drugs that are effluxed and loose efficacy in hypoxic cells.

The high number of CSCs in hypoxic tumor areas [86,87,88] is also associated with chemoresistance, because CSCs are more refractory to chemotherapy, radiotherapy, and immunotherapy than differentiated cells [160]. This resistance is partially due to the overexpression of Pgp [161,162,163] and other transporters of the same family, such as ABCB5 [164]. In addition, the stabilization of HIF-1α in hypoxic CSCs may upregulate several genes related to a highly metastatic and poorly chemosensitive phenotype, such as MMP9, C-X-C chemokine receptor type 4 (CXCR4), osteopontin, IL-8, and VEGF [165,166]. Inhibitors of HIF-1α, such as propofol [167] or tanshinone IIA [168], reversed at the same time EMT and resistance to docetaxel and doxorubicin, confirming the linkage between hypoxia, EMT, and chemoresistance. Recently, an additional mechanism—based on a crosstalk between CAFs and cancer cells—has been described to link chemoresistance and stemness: indeed, the production of TGF-β2, which typically occurs in an acidic TME [81], from CAFs, increases the intra-tumor expression of GLI2 that favors stemness maintenance in colon cancer and chemoresistance [169]. The endogenous activity of HIF-1α in CSCs synergizes with the TGF-β2 produced by CAFs in increasing GLI2 production [169], suggesting that both hypoxia and acidosis cooperate in inducing chemoresistance of CSC component.

At the same time, HIF-1α reduces the efficacy of chemotherapeutic drugs damaging DNA by upregulating topoisomerase 2A, which reseals DNA double strand breaks [170], by activating DNA repair machinery enzymes, such as DNA-protein kinases (DNA-PKs), Ku70, and Ku80 [171], and by downregulating TP53 [172]. The inhibition of TP53 and the consequent resistance to pro-apoptotic stimuli has also been ascribed to HIF-2α [173].

As already mentioned, ROS are increased in hypoxic tumor areas because of the OXPHOS dysfunction. The increase in ROS below sub-cytotoxic levels triggers an adaptive response that limits oxidative damages. One ROS-sensitive factor coordinating this response is the Nuclear factor erythroid 2-related factor 2 (Nrf2), which upregulates anti-oxidant enzymes and MRP1 [174]. These two events contribute to chemoresistance, because the overexpression of anti-oxidant enzymes involved in ROS detoxification [175] also neutralize the ROS produced by chemotherapeutic drugs [176,177,178]. Moreover, MRP1 reduces the intracellular retention of chemotherapeutic agents [179].

The metabolic reprogramming induced by hypoxia must also be taken into account as a key player in inducing cancer chemoresistance.

## 5. The Metabolic Rewiring Occurring in Hypoxic Tumors Supports Chemoresistance

The metabolic rewiring in both glycolysis and mitochondria induced by hypoxia cause chemoresistance by cooperating with enhanced pro-survival pathways and reduced apoptosis, EMT activation, increased DNA repair, alterations in drug metabolism, changes in drug targets [180,181,182].

The increased acidification of TME produced by the upregulation of glycolytic enzymes and MCT4 is a first reason of chemoresistance. On the one hand, the low extracellular pH (pHe) favors the protonation of weak bases, such as anthracyclines, followed by their inactivation and sequestrations within lysosomes once entered within the cancer cell, a mechanism known as “ion trapping” [183,184]. Second, a typical feature of hypoxic and acidic tumor regions is the increased expression of alkalinizing enzymes. The Na^+^/H^+^ exchanger (NHE) is a typical example of transporter that is up-regulated in response to the acidification: the increased intracellular pH (pHi) produced by its activity creates the optimal conditions for Pgp efflux that is maximally efficient at 7.6-7.8 pH [185]. Indeed, restoring pHi to 7.4–7.2 by blocking NHE reverses the resistance to doxorubicin, a typical Pgp substrate, in colon cancer cells [186]. Other alkalinizing enzymes are the plasma membrane associated CAIX and CAXII that are under the direct transcriptional control of HIF-1α [187,188]. CAXII co-localizes with Pgp in several solid tumors [152] and in particular in the CSC component [188]. Such interaction increases the catalytic activity of Pgp by creating slightly alkaline pH at plasma membrane level [189].

Chemoresistance in hypoxic tumor areas has also been associated with altered mitochondrial metabolism, fusion, fission, and mitophagy [190]. On the one hand, since ABC transporters need a constant supply of ATP, one should expect that hypoxic cells—characterized by lower OXPHOS [131] and higher mitophagy [191]—provide less ATP to ABC transporters, thus, resulting more chemosensitive. Contrarily to these expectations, by up-regulating the Bcl-2/adenovirus E1B 19-kDa interacting protein 3 (BNIP3) [192], an inducer of mitophagy, HIF-1α induces chemoresistance to 5-fluorouracil [193], gemcitabine [176], and cisplatin [177]. Indeed, mitophagy allows an efficient recovery of ATP, reducing equivalents and building blocks, which support chemoresistance by increasing ABC transporters activity, prevent the chemotherapy-induced oxidative stress and repair macromolecules damaged by chemotherapeutic agents. The production of ROS often associated with an altered OXPHOS in hypoxic tumor regions may induce mtDNA damages that further reduces the efficiency of OXPHOS [134]. However, such defective mitochondrial energy metabolism triggers a compensatory response characterized by the upregulation of PGC-1α and PGC-1β, which trigger mitobiogenesis. This mechanism has been proved to induce resistance to cisplatin in non-small cell lung cancer with mtDNA mutations that resulted in a 50% reduction of the NADH:ubiquinone oxidoreductase activity [194]. Overall, the mitochondrial-related parameters (lower OXPHOS and ATP production, higher ROS, increased mitophagy, increased mtDNA mutations) that characterize the hypoxic tumor cells induce chemoresistance.

Moreover, cancer cells often have the ability to exploit both glycolysis and OXPHOS, fueled by glutaminolysis and FAO. In this way, cells shift from one energetic pathway to the other one, according to the glucose availability [68]. This metabolic plasticity allows meeting the increasing requirements of energy and building block, necessary to proliferate, migrate, survive, and stimulate neo-angiogenesis in response to stressing agents as chemotherapeutic drugs [195]. Having a constitutively active glycolysis [8], but also an increased rate of FAO [127] and glutaminolysis [113], HIF-1α-expressing cells exhibit a very high metabolic plasticity that allows a better survival in response to glucose and O_2_ shortage, or chemotherapy. We suggest that the pleiotropic effects of hypoxia on metabolic reprogramming all contribute to chemoresistance by different but cooperating mechanisms.

## 6. Counteracting Hypoxic Metabolic Rewiring: A New Generation of Chemosensitizing Agents

Over the last two decades, an enormous effort has been done to look for efficient treatments counteracting chemoresistance. Since the deregulated cell metabolism could cause chemoresistance, specific metabolic modifiers may be potentially effective as chemosensitizer agents.

The most direct approach consists in targeting HIF-1α, given its central role in controlling metabolic reprogramming, proliferation, invasion, and chemoresistance. Inhibitors of HIF-1α translation (EZN-29-68, PX-478, Vorinostat, digoxin), compounds decreasing HIF-1α stability (PX-478, Vorinostat, YC-1) and transcriptional activity (YC-1, acriflavine), are the most efficient inhibitors of HIF-1α [196,197,198,199]. Besides inhibiting directly tumor growth, some inhibitors of HIF-1α transcriptional program, such as YC-1 [155], BAY87-2243 [200], dutasteride [201], and emetine [202] have shown chemosensitizing effects in solid and hematologic tumors at preclinical levels. However, the inhibition of physiological processes controlled by HIF-1α in non-transformed tissues has strongly limited the positive results in patients (https://clinicaltrials.gov/ct2/results/details?cond=Cancer&term=HIF).

Targeting the metabolic pathways modulated by HIF-1α and involved in chemoresistance could represent a second chemosensitizing approach (Figure 3).

Cancer cells highly dependent on glycolysis display a dramatic decrease in ATP upon treatment with the synthetic glucose analog 2-deoxy-D-glucose (2-DG), which blocks the glycolytic flux [203]. Besides exerting strong antitumor effects when used alone, it enhances the effects of chemotherapeutic agents, such as doxorubicin [204]. Although the same objection raised on HIF-1α inhibitors, i.e., the low specificity and the toxicity on non-transformed tissues, can be applied to glycolysis inhibitors, the different dependence from glycolysis between tumor and non-tumor cells make the former more susceptible to 2-DG cytotoxicity, used at dosages minimally affecting normal cells. In addition, physiologically produced metabolites, such as melatonin, have revealed an unexpected effect as anticancer agents, displaying a lower toxicity that synthetic glycolytic inhibitors. Indeed, due to the ability of downregulating GLUT1 and PDK1, melatonin reduces the glycolytic flux and removes the inhibition on PDH exerted by PDK1. This rewiring counteracts the energy production dependent on anaerobic glycolysis [205]. Of note, cancer cells have a lower ability to synthesize melatonin and are more sensitive to the melatonin’s cytotoxic effects than non-transformed cells [205]: this feature can be considered a protective mechanism to maintain ATP production. On the other hand, it suggests that exogenously administered melatonin (that have few side effects on non-transformed cells) can represent a tumor-selective anti-cancer and chemosensitizing strategy.

A chemosensitizing effect of 2-DG in combination with 5-fluorouracil was reported in vitro and in hepatocellular carcinoma xenografts [206]. However, escape mechanisms towards metabolic modifiers have also been reported: indeed, the continuous inhibition of glucose metabolism in tumors treated with 2-DG selects glycolytic-independent clones that result insensitive to 2-DG and appear more aggressive, thanks to the activation of compensatory pro-survival pathways [206].

The 3-bromo-pyruvate (3BrPyr) is a potent inhibitor of HK isoform II, one of the pacemaker enzyme of glycolysis, often overexpressed in tumors; therefore, it induces an ATP crisis. Being an alkylating agent on HKII and other enzymes, including anti-oxidant enzymes, it has been proposed as a potential enhancer of alkylating chemotherapeutic drugs. Indeed, 3BrPyr successfully increases the sensitivity to carmustine in hepatocellular carcinoma cells, by inhibiting glycolysis and inactivating several targeting proteins involved in redox homeostasis [207]. Both HKII and PDK1, two glycolytic enzymes upregulated by HIF-1α but inhibited by wild-type TP53, cooperate in inducing chemoresistance to cisplatin in ovarian cancer, by maintaining high levels of glycolytic flux and pro-survival pathways: indeed, PDK1 activates Akt signaling that counteracts cisplatin cytotoxic effects [208]. The presence of mutated TP53 and/or the constitutively activated HIF-1α enhance the chemoresistance; to the contrary, the simultaneous inhibition of HKII and PDK1 restores cisplatin sensitivity by reducing energetic metabolism and decreasing Akt signaling [208].

PFK binding protein 3 (PFKBP3) is another critical regulator of glycolysis, because it can stimulate or inhibit PFK-2: recently, the selective PFKBP3 inhibitor PFK158 has shown strong anti-cancer properties, as well as chemosensitizing effects in ovarian cancers resistant to carboplatin and docetaxel [209]. PFKBP3 appears a moonlight enzyme: besides regulating glycolysis, it prevents lipid catabolism and preserve lipid droplets in cancer cells. By inhibiting the ATP production via glycolysis, reducing lipid droplets and inducing lipophagy [209], PFK158 limits at least two energy sources for tumors, and reduces the lipid droplet-dependent metastases [81].

Moreover, LDH-A, the last pace-maker glycolytic enzyme and another HIF-1α target gene, has been correlated with chemoresistance: indeed, it is over-expressed in taxol-resistant TNBC, but its inhibition with oxamate restores chemosensitivity by inducing apoptosis consequent to the reduced ATP supply [210]. Notably, chemoresistant cells were more susceptible to the pro-apoptotic effect of oxamate than the chemosensitive counterpart [210], unveiling a metabolic “Achilles’ heel”.

Since the TME acidification typical of hypoxic areas up-regulates alkalinizing proteins that contribute to chemoresistance [186,189], targeting these proteins has been also experimented as chemosensitizing strategy. The NHE inhibitor ethyl-isopropyl amiloride (EIPA) [186], small molecules [152,187,211] or monoclonal antibodies [212] inhibiting CAXII all successfully reverse resistance to Pgp substrates, such as anthracycline and temozolomide. Another approach to reduce acidosis is blocking the H^+^/lactate symporters. MCT1 inhibitors, such as α-cyano-4-hydroxycinnamate (CHC), AR-C155858 and AZD3965, have been shown to reduce tumor growth and progression [67,213], with limited toxicity in vivo. The main limitation of MCT1 inhibitors, however, is the low specificity. For instance, AR-C155858 has a strong suppressive activity against CD8^+^ T-lymphocytes, which need an active MCT1 to preserve their cytotoxic functions [96]. Moreover, blocking MCT1 force cell to increase the rate of glucose oxidation via the TCA cycle and OXPHOS, cancelling the desired anticancer effect of MCT1 inhibitors. For this reason, inhibitors of the mitochondrial pyruvate carrier (MPC) such as 7ACC2, which prevents the entry of pyruvate into TCA/OXPHOS flux, result more effective in reducing tumor growth [214]. Besides producing an ATP crash, the inhibition of MPC has the advantage of reducing the O_2_ consumption by mitochondria, relieving the intratumoral hypoxia, allowing radiosensitization [214] and likely chemosensitization. Finally, it must be considered that the presence of both MCT1 and MCT4 within different tumor areas may limit the efficacy of MCT inhibitors. Indeed, the block of one isoform often induces the compensatory up-regulation of the other isoform [213], preserving the efflux of lactate and the acidification of TME. The simultaneous use of inhibitors of MCT1 and MCT4, such as AZ93, may increase the efficacy of this approach [213], by promoting the shift of cancer cells from a glycolytic-based metabolism toward a OXPHOS-based metabolism. At the moment, however, the presence of severe side toxicity of MPC or MCT inhibitors are the major limitations of their use.

Notwithstanding possible toxicities, the shift from a glycolytic to an OXPHOS-based metabolism strongly kill hypoxic cancer cells that rely on glycolysis for their energy production. The glycolysis to OXPHOS shift is achieved for instance by the PDK1 inhibitor dichloroacetate (DCA) that relieves the inhibition on PDH, allowing the generation of AcCoA and fueling TCA cycle [215]. In these conditions, the defective OXPHOS of hypoxic tumors, not trained to have an efficient ETC, may increase the rate of mistakes, e.g., the amount of uncompleted reduction of O_2_. In turns, the increase in ROS may induce cell death and synergize with chemotherapeutic agents producing ROS, such as cisplatin. On the other hand, the increased amount of ATP derived from OXPHOS, although not sufficient to protect from apoptosis, reduces the glycolytic flux with a feedback inhibition on PFK-2, completely counteracting the metabolic reprogramming induced by hypoxia. Moreover, a direct inhibition of DCA on HIF-1α activity has been documented: this inhibition further contributes to chemosensitization via the direct downregulation of Pgp [215]. The supplementation of doxorubicin-resistant colon cancer and malignant pleural mesothelioma cells, constitutively expressing HIF-1α, with pyruvate restores the sensitivity to doxorubicin: the mechanisms is similar to that of DCA, because pyruvate increases ETC and suppresses glycolysis [216], reversing the metabolic phenotype supported by HIF-1α. The increased OXPHOS and the consequent increase of mtROS may explain the chemosensitization to doxorubicin.

Dual targeting compounds, i.e., agents reprogramming at least two metabolic pathways as DCA and pyruvate, are the most effective chemosensitizing strategies among the metabolic modifiers. In etoposide-resistant neuroblastoma, a constitutively high level of anti-apoptotic survivins induce mitochondria fission, downregulate OXPHOS and increase glycolysis: this signature, similar to that produced by HIF-1α, is associated to a metastatic and chemoresistant phenotype. By relying mainly on glycolysis to obtain ATP, resistant neuroblastoma cells are highly susceptible to glycolytic inhibitors: indeed, the treatment with 2-DG reprograms the metabolic phenotype of resistant neuroblastoma and at the same time sensitizes cells to etoposide [217]. The combination of two different metabolic modifiers also achieve a wide reprogramming: for instance, the mitochondria-targeted carboxy-proxyl nitroxide (mito-CP) compound inhibits mitochondrial metabolism, increases the levels of citrate that cannot be metabolized in the TCA cycle, but it is exported in the cytosol, and inhibits glycolysis at the PFK-2 step. In association with 2-DG, mito-CP strongly induces apoptosis in pancreatic adenocarcinoma [218], one of the most chemoresistant tumor, demonstrating that the combined use of metabolic modifiers may be effective against refractory cancers. Similarly, the simultaneous inhibition of glycolysis with 2-DG or lonidamine, and of FAO with etomoxir, enhances the anti-cancer effect of arsenic trioxide (AsO) in acute pro-myelocytic leukemia cells [219]: the mechanism is related to a significant drop in ATP levels and to a down-regulation of the pro-survival ERK1/2 and Akt-dependent pathways. Although no toxic effects were reported on peripheral blood monocytes in vitro [219], a severe toxicity of combination therapies cannot be excluded in vivo, considering that glycolysis and FAO, i.e., the two main energy pathways for most tissues, are blocked. Similarly, combination therapies based on MCT inhibitors and OXPHOS inhibitors [213], may represent an effectively lethal tool against chemoresistant cells in hypoxic tumor regions. Also in this case, the potential toxicity represents the main limitation, since these combinatorial approaches induce a metabolic catastrophe.

Since hypoxia increases FA synthesis, agents targeting this pathway may contribute to chemosensitization. For instance, the FASN inhibitor Orlistat induces chemosensitization to cisplatin in T-cell lymphoma, by pleiotropic mechanisms, including increased ROS levels, decreased MCT1 expression and lactate efflux, downregulated Pgp and MRP1 [220]. Although the linkages connecting all these events and the levels of HIF-1α have not been evaluated in this study, it may pave the way to consider the targeting of FA metabolism as a new chemosensitizing approach.

Several small molecules altering mitochondria metabolism, termed “mitocans”, have shown promising anti-cancer efficacy [221], in particular in tumors where glycolysis has a minor role in producing ATP. Since mitochondrial metabolism is often attenuated in hypoxic tumors, it could be further deranged by mitocans, to produce a deep energetic crash, induce apoptosis and chemosensitization [222]. The potential across the inner membrane of mitochondria is highly negative inside (~ −180 mV). This difference allows lipophilic cations, such as triphenyl alkyl phosphonium (TPP^+^) to accumulate preferentially within mitochondria [221]. Phenol TPP^+^-derivatives have recently shown to lower mitochondrial biogenesis, mitochondrial mass and energetic metabolism, increasing at the same time the mtROS levels, opening the membrane permeability transition pore, and triggering a caspase-9-mediated apoptosis [223]. By reducing OXPHOS and increasing mtROS, TPP^+^-derivatives may worsen the energetic mitochondrial metabolism of hypoxic cancer cells and increase at the same time the possible oxidative damages. They could be exploited as chemosensitizers in hypoxic areas of tumors, in combination with chemotherapeutic drugs that generate ROS, such as cisplatin, gemcitabine, or doxorubicin [176,177,178,179]. The TPP^+^-conjugate of DCA (mito-DCA) is a synthetic derivative of DCA, conceived as a more potent version of parental compound: the vectorization to mitochondria amplifies the inhibition of PDK1 and the consequent shift from glycolysis to OXPHOS [224]. This property can be exploited in hypoxic cells that heavily rely on glycolysis for their energy requirements. The anti-diabetic drug metformin has been repurposed as a potential anti-cancer drug, as it inhibits complex I of ETC [225]. Various mitochondria-targeted metformin analogues, with different alkyl chain lengths, have been synthesized, with the aim of strengthening the inhibition of OXPHOS. The anti-proliferative effects of mito-metformin increase with the increasing length of alkyl chain [226]. As a consequence of the strong inhibition of OXPHOS, these mito-metformin increase the production of ^•^O2^−^ and H_2_O_2_, and are radio-sensitizers in pancreatic cancer cells [226]. Thanks to their ability of increasing mtROS, they could be potential chemosensitizer agents. Similarly, the increase of mtROS in chemoresistant cells constitutively expressing HIF-1α has been achieved by the metronomic administration of doxorubicin: two consecutive low doses of the drug increases the ETC flux but also the mtROS that damage the mitochondrial membrane, decrease the yield of ATP and trigger apoptosis [227], fully restoring chemosensitivity. The increase in mtROS is not the only mechanism explaining the anti-cancer efficacy of OXPHOS inhibitors. For instance, the complex I inhibitor IACS-010759 shows good anti-cancer activity because of the marked reduction in energy production and in aspartate regeneration via TCA-dependent cataplerotic/anaplerotic pathways. The low ATP synthesis and the reduced pyrimidine synthesis caused by the lower aspartate levels explain the anti-cancer efficacy of IACS-010759, currently approved in phase I trials for acute myeloid leukemia and glioblastoma [228]. IACS-010759 has overcome the common limitations observed by other OXPHOS targeting agents, such as off-target effects and unfavorable pharmacokinetic profile [228], showing acceptable toxicity in non-transformed cells if used in the proper therapeutic window. Notably, IACS-010759 belongs to a library of compounds with high specificity for complex I and for cells relying on OXPHOS to produce ATP [228]. For this reason, the compound, which does not affect HIF-1α transcriptional program, is effective in glucose-deprived hypoxic cells [229]. Moreover, it can be considered a good metabolic modifier in acidosis that downregulates HIF-1α transcriptional program and glycolytic enzymes [83], forcing cancer cells to rely mainly on OXPHOS to meet their demand of ATP.

Overall, these pharmacological evidences demonstrate that shifting the glycolysis/OXPHOS balance may be an effective chemosensitizing strategy against hypoxic tumor cells.

## 7. Conclusions and Future Perspectives

This review provides a deep analysis of the linkages between hypoxia, metabolic rewiring, and chemoresistance in cancers. Drug resistance is well known as the primary cause of therapeutic failure in cancer treatment. The mechanisms of chemoresistance involve a combination of cell-intrinsic factors such as oncogenic drivers or mutations, TME-associated factors, pharmacokinetic factors. Hypoxia is a common feature of TME. It plays an important role in selecting cells challenged by a low O_2_ supply and forced to rewire their metabolism. This training to survive under unfavorable conditions inevitably make cells more resistant to exogenous stresses such as chemotherapy. HIF-1α-dependent chemoresistance relies on the transcriptional activation of genes determining cell proliferation, metastasis, EMT, maintenance of stem cell-like properties, drug efflux, and metabolic reprogramming.

The typical metabolic signature of hypoxic tumors is characterized by increased glucose uptake and fermentation into lactate, decreased pHe/increased pHi, reduced TCA cycle and OXPHOS, increased production of mtROS, increased uptake of AAs, and increased synthesis of anti-oxidant metabolites as GSH. All of these features contribute to chemoresistance. The accelerated glycolysis supplies cells with sufficient ATP to promote cell survival and with glycolysis intermediates for biosynthetic purposes. The increased mitophagy characterizing hypoxic cells sustains the possibility of recovering ATP. The altered pH inactivates several drugs that are active as weak bases, or sequesters them into lysosomes after protonation. The increased production of sub-cytotoxic mtROS, coupled with the higher levels of anti-oxidant metabolites, train cells to be less susceptible to the oxidative damage induced by chemotherapy.

If targeting HIF-1α could represent an effective approach because it blocks a great number of processes determining chemoresistance, HIF-1α inhibitors have the disadvantage of huge toxicity, due to the inhibition of physiologically important HIF-1α-dependent processes, such as ischemia-reperfusion response in non-transformed tissues. Targeting the metabolic pathways controlled by HIF-1α and re-programming them as in normoxic cells may improve the efficacy of chemotherapeutic drugs and/or attenuate chemoresistance. Although metabolic modifiers have already been tested in clinical trials, side effects deriving from the inhibition of metabolic pathways in non-transformed tissues cannot be excluded. However, the metabolic signature of hypoxic cancer cells—based on high anaerobic glycolysis and low mitochondrial metabolism—is quantitatively different from the normal tissues. This quantitative difference may open a therapeutic window at which metabolic modifiers could be safely used against hypoxic cancer cells, without damaging non-transformed tissues. Nanotechnology-based drug delivery, employing tumor-targeting liposomes or nanoparticles, could aid to increase the delivery and the vectorization of the metabolic modifiers towards the tumor, increasing the therapeutic benefits and reducing the side effects. Finally, some endogenous metabolites, such as pyruvate or melatonin, differentially produced by hypoxic cells and non-tumor tissues, have revealed significant chemosensitizing properties, coupled with lower risks of toxicity.

Thanks to the deep knowledge of the metabolic rewiring induced by hypoxia, and causing chemoresistance, a precision medicine based on specific metabolic modifiers can be proposed as a novel chemosensitizing strategy against aggressive and refractory tumors.

## Figures and Tables

**Figure 1 cells-09-02598-f001:**
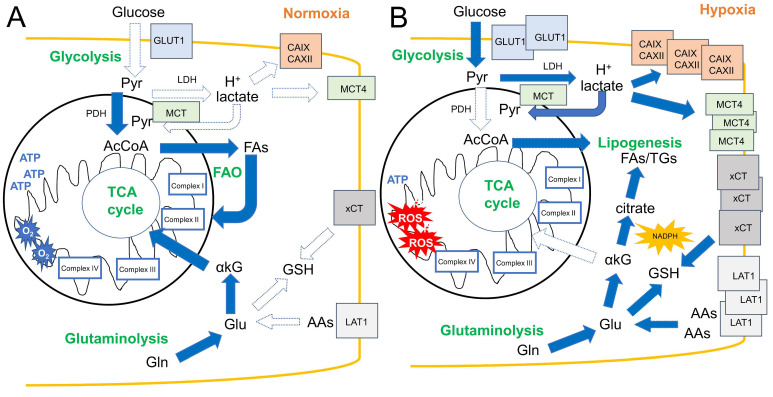
Metabolic rewiring induced by hypoxia. (**A**) In normoxic tumor areas, glucose taken up by glucose transporter 1 (GLUT1) and transformed by glycolysis into pyruvate is mainly oxidized into Acetyl-Coenzyme A (AcCoA) by pyruvate dehydrogenase (PDH), fueling the tricarboxylic acid (TCA) cycle and the electron transport chain to produce ATP. By contrast, lactic fermentation is low. Fatty acid (FAs) oxidation (FAO) prevails over FA and triglyceride (TG) synthesis, further producing AcCoA entering the TCA cycle. The oxidative metabolism of glutamine (Gln) is a third anaplerotic pathway of TCA cycle. (**B**) Hypoxia enhances glucose uptake via GLUT1 and metabolism into lactate, thanks to the hypoxia-inducible factor (HIF)-1α-dependent upregulation of glycolytic enzymes, lactate dehydrogenase (LDH), and pyruvate dehydrogenase kinase 1 that inhibits PDH. In this way, cancer cells produce huge amounts of lactate, exported by the hypoxia-sensitive gene mono-carboxylate transporter 4 (MCT4) and H^+^, buffered by other HIF-1α-targeted genes, such as carbonic anhydrase (CA) IX and XII. Part of lactate can enter the mitochondria via MCT, and is transformed into pyruvate by mitochondrial LDH, fueling TCA cycle. Notwithstanding such anaplerosis, TCA and electron transport chain (ETC) are reduced in hypoxic tumors that aberrantly reduce O_2_ into reactive oxygen species (ROS). Amino acids (AAs) uptake is increased thanks to the upregulation of xCT and L-type amino acid transporter 1 (LAT1), also induced by HIF-1α. The catabolism of Gln and glutamate (Glu), however, not only fuels TCA cycle, since it is mainly used to synthesize glutathione (GSH) or produce α-ketoglutarate (αKG) that is reduced into citrate and NADPH via the reductive glutamine metabolism, providing carbons for FA and triglycerides (TGs) synthesis. Dotted arrows: down-modulated metabolic pathways; solid arrows: up-regulated metabolic pathways.

**Figure 2 cells-09-02598-f002:**
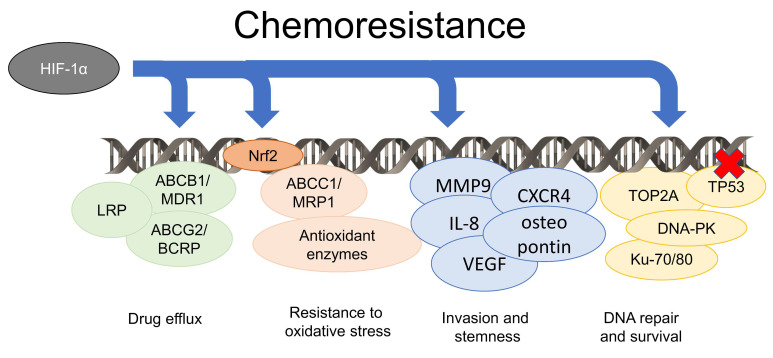
Metabolism-independent factors determining chemoresistance in hypoxia. HIF-1α transcriptionally induces drug efflux transporters belonging to the adenosine triphosphate (ATP) Binding Cassette (ABC) family, such as ABC transporter B1/multidrug resistance 1 (ABCB1/MDR1, known as P-glycoprotein/Pgp), ABC transporter G2/breast cancer resistance protein (ABCG2/BCRP), lung cancer resistance protein (LRP), ABC transporter C1/multi-drugs resistance related protein 1 (ABCC1/MRP1), by cooperating with the transcription factor Nuclear factor erythroid 2-related factor 2 (Nrf2) that also up-regulates anti-oxidant enzymes. Moreover, HIF-1α increases genes that induce the acquisition of a pro-invasive and stem-like phenotype, such as matrix metalloprotease 9 (MMP9), C-X-C chemokine receptor type 4 (CXCR4), osteopontin, interleukin-8 (IL-8), and vascular endothelial growth factor (VEGF), and up-regulates genes promoting DNA repair, such as topoisomerase 2A (TOP2A), DNA-protein kinases (DNA-PKs), Ku-70, and Ku-80, inhibits TP53 activity, preventing DNA damage. This complex and coordinated transcriptional program favors cell survival and determines chemoresistance in cancer cells.

**Figure 3 cells-09-02598-f003:**
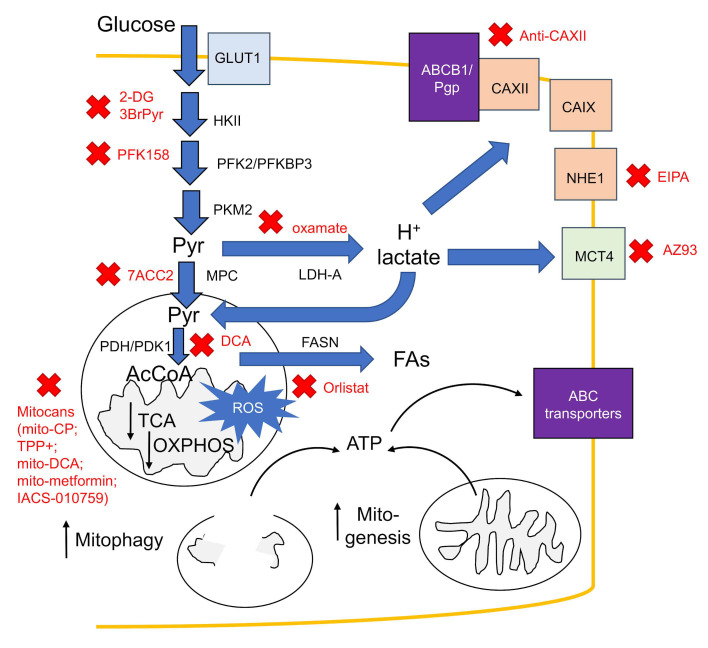
Metabolism-dependent factors inducing chemoresistance in hypoxia and possible therapeutic strategies. In hypoxic tumor areas, HIF-1α increases glucose uptake and anaerobic glycolysis, thanks to the upregulation of glucose transporter 1 (GLUT1), hexokinase II (HKII), phosphofructokinase 2/phosphofructokinase binding protein 3 (PFK2/PFKBP3), pyruvate kinase M2 (PKM2) and lactate dehydrogenase A (LDH-A), with consequent export of lactate and H^+^ via monocarboxylate transporter 4 (MCT4). The decreased extracellular pH inactivates chemotherapeutic drugs, such as anthracyclines, and is compensated by the Na^+^/H^+^ exchanger (NHE) and by the carbonic anhydrase (CA) IX and XII. These enzymes create a slightly alkaline pH in the plasma membrane, which is optimal for the catalytic efficiency of ATP binding cassette transporter B1/P-glycoprotein (ABCB1/Pgp). Part of lactate may enter in the mitochondria via MCT and be transformed into pyruvate by a mitochondrial LDH isoform. Pyruvate is imported in mitochondria by the mitochondrial pyruvate carrier (MPC), but the low activity of pyruvate dehydrogenase (PDH) produced by the HIF-1α-driven upregulation of pyruvate dehydrogenase kinase 1 (PDK1) and the consume of Acetyl Coenzyme A (AcCoA) by fatty acid (FA) synthase (FASN) reduce the metabolic flow through the tricarboxylic acid (TCA) cycle and the oxidative phosphorylation (OXPHOS). The reduced mitochondrial metabolism, coupled with increased mitophagy and compensatory mitogenesis promoted by HIF-1α, results in paradoxically high levels of ATP, which supports the catalytic activity of ABC transporters, increasing the efflux of multiple chemotherapeutic drugs. This wide metabolic rewiring induces chemoresistance but it can be counteracted by metabolic modifiers inhibiting specific enzymes (indicated in red). Abbreviations: 2-DG: 2-deoxy-D-glucose; 3BrPyr: 3-Bromo-pyruvate; EIPA: ethyl-isopropyl amiloride; DCA: dichloroacetate; mito-CP: mitochondria-targeted carboxy-proxyl nitroxide; TPP^+^: triphenyl alkyl phosphonium; mito-DCA: mitochondria-targeted dichloroacetate; mito-metformin: mitochondria-targeted metformin.

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
