# Peer review of "Hypoxia Dictates Metabolic Rewiring of Tumors: Implications for Chemoresistance"

_cells, 2020, doi:10.3390/cells9122598_

Round 1

Reviewer 1 Report

Dear Editor, dear Authors,

the paper entitled “Hypoxia dictates metabolic rewiring of tumors: implications for chemoresistance” by Dimas Carolina Belisario, Joanna Kopecka, Martina Pasino, Muhlis Akman, Enrico De Smaele, Massimo Donadelli and Chiara Riganti, recapitulates the mechanisms by which HIF-a (HIF-1a or HIF-2a), the O2-regulated subunit of HIF, is activated (paragraph 2), the main targets of active HIF dimer and the effect of this transcriptional inducer (and of hypoxia in general) on cancer metabolic rewiring (paragraph 3). The authors also describe some of the mechanisms by which HIF favors a metabolic rewiring leading to highly metastatic and chemoresistant cancer phenotype (paragraph 4). Finally, the molecular mechanisms of metabolic reprogramming that induce chemoresistance (paragraph 5), an argument already described more in detail in a recent review paper by the same corresponding author [Teodora Alexa-Stratulata, Milica Pešić, Ana Čipak Gašparović, Ioannis P. Trougakos, Chiara Riganti (2019) What sustains the multidrug resistance phenotype beyond ABC efflux transporters? Looking beyond the tip of the iceberg. Drug Resistance Updates, 46, 100643] (ref. n. 155 of the present manuscript), and the therapeutic strategies currently adopted to counter drug resistance are reviewed.

The manuscript is well written and provides a good and concise overview of the topics covered, even if several similar reviews have been published, dealing on HIF, hypoxia and their effects on cancer cell metabolism, chemoresistance and metastatic capacity [Masson, N., Ratcliffe, P.J. 2014 Cancer Metab.; Hong Xie and M. Celeste Simon 2017 JBC; Front Cell Dev Biol. 2019; Teodora Alexa-Stratulata 2019 Drug Resistance Updates; Wafaa Al Tameemi et al. 2019 Front Cell Dev Biol.; Wei, Q. et al. 2020 Oncogene]. The manuscript also provides a quite comprehensive list of relevant references.

A revision of the manuscript according to the issues listed below is required to improve it and possibly to differentiate it from other similar published review articles:

  1. A description of the mechanisms by which HIF/hypoxia might induce the shift from glutamine oxidation to reductive glutamine metabolism (needed for fatty acid synthesis, cholesterol synthesis and the maintenance of oxidative homeostasis via NADPH production under hypoxic conditions) is not provided (Paragraph 3.2). Moreover, that serine catabolism can contribute, together with glutamine, to lipid synthesis and regulate redox control during hypoxia via serine hydroxymethyltransferase which contributes to NADPH production [Porporato PE et al 2018 Cell Research], should be mentioned.

  1. MCTs overexpression and increased activity are common features of cancers. Their importance relies on their ability to transport lactate, the real final product of glycolysis [Schurr A 2006 J Cereb Blood Flow Metab.; Quistorff B and Grunnet N 2011 Aging]. Lactate plays a role in cancer energy metabolism and anabolism [Lemire J et al. 2008 PLoS One; Chen YJ, Mahieu NG, Huang X et al 2016 Nat Chem Biol; de Bari and Atlante 2019 Cell Mol Life Sci], as well as in tumor growth, angiogenesis, oxidative stress regulation and cell signaling [Brooks GA 2016 Adv Exp Med Biol; San-Millán I 2017 Carcinogenesis; Brooks GA 2018 Cell Metabolism; de Bari and Atlante 2019 Cell Mol Life Sci]. Lactate is also involved in cell–cell interaction: a metabolic symbiosis exists in cancer based on the MCT-mediated transfer of glycolysis-derived lactate from hypoxic to normoxic areas, where it would be employed to fuel OXPHOS as a strategy to avoid competition for glucose on which glycolytic cancer cells strictly rely, this explaining why targeting lactate-fueled respiration selectively kills hypoxic tumor cells [Sonveaux P et al. 2008 J Clin Invest]. Finally, lactate can activate HIF-1 and increases MCT1 expression [see Brooks GA 2018 Cell Metabolism]. In the present paper little emphasis is placed on the interconnection between MCTs, lactate and HIF in cancer metabolic rewire. A brief paragraph dealing with these topics and the therapeutic strategies targeting MCTs and/or lactate synthesis and metabolism should be added. This would also make the difference between this review and the above mentioned ones dealing with HIF.

  1. Lactate is known to get out of glycolytic cancer cells mainly via MCT4, while it is taken up by oxidative cancer cells via MCT1 [Ullah MS, Davies AJ, Halestrap AP 2006 J Biol Chem; Sonveaux P et al. 2008 J Clin Invest]. Why do the authors report MCT1 as lactate carrier in hypoxic tumors (Fig. 1 and Fig. 3)?

  1. Paragraph 3.4: The authors state that mitochondria are “dangerous” organelles for hypoxic tumors and “mitochondrial dysfunction” occurs. Since targeting mitochondrial function and OXPHOS affects cell survival even in hypoxic cancers (as also described in Paragraph 6.), it is feasible that mitochondria are not damaged/dysfunctional in these cells, but likely somehow useful to them even under hypoxic conditions. Then, “reduced mitochondrial metabolism” (as stated by the authors in the Abstract) appears to be more correct.

  1. Paragraph 6: Check the description of melatonin effects (lines 495-500). The authors state that “due to the ability of down-regulating GLUT1 and PDK1, melatonin reduces the consequent glycolysis and the transformation of pyruvate in AcCoA, reducing both anaerobic and aerobic energy metabolism [170]”. On the contrary, as reported in ref. n. 170 (Reiter RG et al 2020 Cell. Mol. Life Sci), melatonin inhibits GLUT1 and PDK1, thus reducing the inhibitory effect of PDK on pyruvate dehydrogenase complex and allowing for the restoration of pyruvate metabolism by mitochondria, reducing the Warburg effect.

Author Response

Dear Editor, dear Authors,

the paper entitled “Hypoxia dictates metabolic rewiring of tumors: implications for chemoresistance” by Dimas Carolina Belisario, Joanna Kopecka, Martina Pasino, Muhlis Akman, Enrico De Smaele, Massimo Donadelli and Chiara Riganti, recapitulates the mechanisms by which HIF-a (HIF-1a or HIF-2a), the O2-regulated subunit of HIF, is activated (paragraph 2), the main targets of active HIF dimer and the effect of this transcriptional inducer (and of hypoxia in general) on cancer metabolic rewiring (paragraph 3). The authors also describe some of the mechanisms by which HIF favors a metabolic rewiring leading to highly metastatic and chemoresistant cancer phenotype (paragraph 4). Finally, the molecular mechanisms of metabolic reprogramming that induce chemoresistance (paragraph 5), an argument already described more in detail in a recent review paper by the same corresponding author [Teodora Alexa-Stratulata, Milica Pešić, Ana Čipak Gašparović, Ioannis P. Trougakos, Chiara Riganti (2019) What sustains the multidrug resistance phenotype beyond ABC efflux transporters? Looking beyond the tip of the iceberg. Drug Resistance Updates, 46, 100643] (ref. n. 155 of the present manuscript), and the therapeutic strategies currently adopted to counter drug resistance are reviewed.

The manuscript is well written and provides a good and concise overview of the topics covered, even if several similar reviews have been published, dealing on HIF, hypoxia and their effects on cancer cell metabolism, chemoresistance and metastatic capacity [Masson, N., Ratcliffe, P.J. 2014 Cancer Metab.; Hong Xie and M. Celeste Simon 2017 JBC; Front Cell Dev Biol. 2019; Teodora Alexa-Stratulata 2019 Drug Resistance Updates; Wafaa Al Tameemi et al. 2019 Front Cell Dev Biol.; Wei, Q. et al. 2020 Oncogene]. The manuscript also provides a quite comprehensive list of relevant references.

A revision of the manuscript according to the issues listed below is required to improve it and possibly to differentiate it from other similar published review articles:

1. A description of the mechanisms by which HIF/hypoxia might induce the shift from glutamine oxidation to reductive glutamine metabolism (needed for fatty acid synthesis, cholesterol synthesis and the maintenance of oxidative homeostasis via NADPH production under hypoxic conditions) is not provided (Paragraph 3.2). Moreover, that serine catabolism can contribute, together with glutamine, to lipid synthesis and regulate redox control during hypoxia via serine hydroxymethyltransferase which contributes to NADPH production [Porporato PE et al 2018 Cell Research], should be mentioned.

We recognize that the mechanisms linking hypoxia and reductive glutamine catabolism were not adequately explained. We detailed them in paragraph 3.3. of the new version (line 321) and added two new references. We modified figure 1 and its legend. Furthermore, as suggested by the Reviewer, we mentioned that also serine catabolism may contribute to produce NADPH via the HIF-2-induced up-regulation of serine hydroxymethyltransferase 2 enzyme (line 335). We added one new reference and anticipated the reference [Porporato PE et al 2018 Cell Research] in the revised version.

2. MCTs overexpression and increased activity are common features of cancers. Their importance relies on their ability to transport lactate, the real final product of glycolysis [Schurr A 2006 J Cereb Blood Flow Metab.; Quistorff B and Grunnet N 2011 Aging]. Lactate plays a role in cancer energy metabolism and anabolism [Lemire J et al. 2008 PLoS One; Chen YJ, Mahieu NG, Huang X et al 2016 Nat Chem Biol; de Bari and Atlante 2019 Cell Mol Life Sci], as well as in tumor growth, angiogenesis, oxidative stress regulation and cell signaling [Brooks GA 2016 Adv Exp Med Biol; San-Millán I 2017 Carcinogenesis; Brooks GA 2018 Cell Metabolism; de Bari and Atlante 2019 Cell Mol Life Sci]. Lactate is also involved in cell–cell interaction: a metabolic symbiosis exists in cancer based on the MCT-mediated transfer of glycolysis-derived lactate from hypoxic to normoxic areas, where it would be employed to fuel OXPHOS as a strategy to avoid competition for glucose on which glycolytic cancer cells strictly rely, this explaining why targeting lactate-fueled respiration selectively kills hypoxic tumor cells [Sonveaux P et al. 2008 J Clin Invest]. Finally, lactate can activate HIF-1 and increases MCT1 expression [see Brooks GA 2018 Cell Metabolism]. In the present paper little emphasis is placed on the interconnection between MCTs, lactate and HIF in cancer metabolic rewire. A brief paragraph dealing with these topics and the therapeutic strategies targeting MCTs and/or lactate synthesis and metabolism should be added. This would also make the difference between this review and the above mentioned ones dealing with HIF.

We thank the Reviewer for the helpful suggestions. We added a dedicated paragraph to the role lactate, MCT1 and MCT4 in hypoxic tumors, and on the implication of the lactate shuttle in tumor growth, invasion and immune-escape (paragraph 3.2 in the revised manuscript; line 173). Moreover, we reported the latest development of MCT inhibitors and discussed advantages and disadvantages of this approach in section 6 (line 635). We modified figures 1 and 3, and their legends. We updated the references section accordingly.

3. Lactate is known to get out of glycolytic cancer cells mainly via MCT4, while it is taken up by oxidative cancer cells via MCT1 [Ullah MS, Davies AJ, Halestrap AP 2006 J Biol Chem; Sonveaux P et al. 2008 J Clin Invest]. Why do the authors report MCT1 as lactate carrier in hypoxic tumors (Fig. 1 and Fig. 3)?

We apologize for the conceptual mistake. We specified the role of MCT1 and MCT4 in the manuscript (line 179), modifying the reference section accordingly. We changed figures 1 and 3, and the respective legends.

4. Paragraph 3.4: The authors state that mitochondria are “dangerous” organelles for hypoxic tumors and “mitochondrial dysfunction” occurs. Since targeting mitochondrial function and OXPHOS affects cell survival even in hypoxic cancers (as also described in Paragraph 6.), it is feasible that mitochondria are not damaged/dysfunctional in these cells, but likely somehow useful to them even under hypoxic conditions. Then, “reduced mitochondrial metabolism” (as stated by the authors in the Abstract) appears to be more correct.

We thank the Reviewer for the observation. We changed the title of the paragraph (3.5 in the new version), the first sentence (line 388) anc corrected misleading sentences throughout the manuscript.

5. Paragraph 6: Check the description of melatonin effects (lines 495-500). The authors state that “due to the ability of down-regulating GLUT1 and PDK1, melatonin reduces the consequent glycolysis and the transformation of pyruvate in AcCoA, reducing both anaerobic and aerobic energy metabolism [170]”. On the contrary, as reported in ref. n. 170 (Reiter RG et al 2020 Cell. Mol. Life Sci), melatonin inhibits GLUT1 and PDK1, thus reducing the inhibitory effect of PDK on pyruvate dehydrogenase complex and allowing for the restoration of pyruvate metabolism by mitochondria, reducing the Warburg effect.

We apologize for reporting incorrect information regarding ref. 170, we corrected the sentence in the revised version (line 594).

Reviewer 2 Report

In the current manuscript, the authors make an overview of the literature on hypoxia-driven metabolic preferences in tumor cells and the interplay with the response to chemotherapeutic drugs.

The manuscript is overall quite well written and organized but it suffers from conceptual flaws.

1-Indeed, throughout the manuscript, the authors use the wording "hypoxic tumors" and even make the comparison with "normoxic tumors", making them "chemoresistant" or "chemosensitive" respectively. This is a very simplistic (and wrong) view. The authors must better explain that it is a matter of spatial compartments within a same tumor, with hypoxic regions that shape a specific cancer cell phenotype (among which metabolism, the focus of the review) and that act as local permissive niches for stem-like relapse-inducing cancer cells (minimal residual disease).

2-the notion of cycling hypoxia must be better explained (from line 57)

3-the figure 1 is not very comprehensive with multiple arrows. The figure must be reedited to better illustrate the notion of spatial compartments. Also, MCT4 is a well-known hypoxia-induced lactate transporter (Ullah et al JBC 2006) and often regulates lactate efflux in hypoxia-exposed cancer cells, while MCT1 mediates lactate influx (in well-oxygenated tumor areas). This must be corrected in the figure.

4-Extracellular acidosis often overlaps with hypoxia within tumors and many papers have now documented acidosis-mediated metabolic reprogramming in tumor cells (Lamonte et al Cancer Metab 2013; Corbet et al Nat Commun 2020...). On the other side, acidosis also induces chemoresistance (via ion trapping, immune surveillance escape, EMT...). The authors must discuss this point and refer to adequate recent literature.

5-The authors must also refer to hypoxia-driven metabolic rewiring in non-cancer cells (immune cells, CAF). This is of particular importance to discuss the broader application of integrating and exploiting hypoxia for resistance to chemotherapy but also immunotherapy, PARP inhibitors or targeted therapies.

6-Besides targeting hypoxia-mediated metabolic pathways, another approach is to inhibit mitochondrial respiration (through MPC inhibition, OXPHOS inhibition with a variety of compounds such as 7ACC2, IACS-010759, metformin...). This must be discussed.

To save space for adding new information in the manuscript, the authors may shorten the sections 1 & 2 which refer to basic and well-described hypoxia-driven molecular mechanisms.

Author Response

In the current manuscript, the authors make an overview of the literature on hypoxia-driven metabolic preferences in tumor cells and the interplay with the response to chemotherapeutic drugs.

The manuscript is overall quite well written and organized but it suffers from conceptual flaws.

1-Indeed, throughout the manuscript, the authors use the wording "hypoxic tumors" and even make the comparison with "normoxic tumors", making them "chemoresistant" or "chemosensitive" respectively. This is a very simplistic (and wrong) view. The authors must better explain that it is a matter of spatial compartments within a same tumor, with hypoxic regions that shape a specific cancer cell phenotype (among which metabolism, the focus of the review) and that act as local permissive niches for stem-like relapse-inducing cancer cells (minimal residual disease).

We apologize for not having clearly explained this point. We agree that within the same tumors there are normoxic and hypoxic areas: such heterogeneity determines metabolic niches that are associated with more chemosensitive or hemoresistant cells. The latter are usually characterized by higher stemness and higher tendency to relapse. We detailed the above-mentioned issue in paragraph 4 (line 454). We added two new references. We corrected the misleading sentences throughout the manuscript and legends.

2-the notion of cycling hypoxia must be better explained (from line 57)

We explained better that the balance between neo-angiogenesis and vessels collapse under the pressure of tumor and stromal cells growth causes fluctuations of O2 levels in opposite directions. This situation produces repeated cycles of hypoxia and normoxia within tumor bulk (line 45).

3-the figure 1 is not very comprehensive with multiple arrows. The figure must be reedited to better illustrate the notion of spatial compartments. Also, MCT4 is a well-known hypoxia-induced lactate transporter (Ullah et al JBC 2006) and often regulates lactate efflux in hypoxia-exposed cancer cells, while MCT1 mediates lactate influx (in well-oxygenated tumor areas). This must be corrected in the figure.

We edited figure 1 and its legend, highlighted the compartmentalization of each pathways. We corrected MCT1 in MCT4.

4-Extracellular acidosis often overlaps with hypoxia within tumors and many papers have now documented acidosis-mediated metabolic reprogramming in tumor cells (Lamonte et al Cancer Metab 2013; Corbet et al Nat Commun 2020...). On the other side, acidosis also induces chemoresistance (via ion trapping, immune surveillance escape, EMT...). The authors must discuss this point and refer to adequate recent literature.

Following the Reviewer’s suggestion, we added a new paragraph (3.2), focused on lactate metabolism and acidosis in hypoxia. We focused on the effect of extracellular acidosis in tumor metabolic reprogramming (line 213), stemness, invasion and EMT (line 242), and immune-evasion (line 273). We updated the references section accordingly. The linkages between hypoxia, stemness and chemoresistance have been discussed in paragraph 4 (lines 478 and 486). The linkages between acidosis and chemoresistance have been discussed in paragraph 5 (line 528). We added two new references in these paragraphs.

5-The authors must also refer to hypoxia-driven metabolic rewiring in non-cancer cells (immune cells, CAF). This is of particular importance to discuss the broader application of integrating and exploiting hypoxia for resistance to chemotherapy but also immunotherapy, PARP inhibitors or targeted therapies.

We thank the Reviewer for the suggestions. We described the effect of hypoxia in CAFs and immune cells infiltrating the TME in the new paragraph 3.2 (lines 192, line 255). We preferred not discussing the implications of such rewiring in resistance to immune-therapy or targeted therapies because our review is focused mainly on chemoresistance and we would not like to go out of focus.

6-Besides targeting hypoxia-mediated metabolic pathways, another approach is to inhibit mitochondrial respiration (through MPC inhibition, OXPHOS inhibition with a variety of compounds such as 7ACC2, IACS-010759, metformin...). This must be discussed.

The strategies targeting mitochondrial metabolism, using different inhibitors of OXPHOS including metformin, have been discussed in paragraph 6 (line 702). We implemented this paragraph, by discussing the potential efficacy and limitations of 7ACC2 (line 642) and IACS-010759 (line 731). We added three new references.  

7-To save space for adding new information in the manuscript, the authors may shorten the sections 1 & 2 which refer to basic and well-described hypoxia-driven molecular mechanisms.

As suggested, we shortened sections 1 and 2.

Round 2

Reviewer 1 Report

Critical issues for the revised Manuscript:

  1. Chapter 3.2: At lines 187 and 189-190 the authors state that L-lactate could be used by cancer cells to produce glucose by gluconeogenesis (“…promoting the synthesis of new glucose [65,73]”; “second, glucose is even regenerated by more oxygenated cells that reach a metabolic self-sufficiency [65,73]”). The ability of cancer cells to generate glucose from lactate, as well as from other non-carbohydrate compounds (glutamine, alanine, and glycerol), is still debated, since there are discrepancies in the effects of gluconeogenic enzymes, such as PEPCK, in different tumors (PEPCK has a tumor suppressing effect in hepatoma and renal cancer, and a protumorigenic role in lung cancer and melanoma) probably due to tissue of origin. It should be also stressed that liver and kidney are the only two organs that express all genes required for a functional gluconeogenic pathway. Other organs as colon or lung cannot produce glucose, but may engage in truncated gluconeogenesis to support their biosynthetic needs rather than to increase glucose production [Wang, Z., & Dong, C. (2018). Gluconeogenesis in Cancer: Function and Regulation of PEPCK, FBPase, and G6Pase. Trends in Cancer. doi:10.1016/j.trecan.2018.11.003]. For these reasons, the above mentioned statements in the revised version of the manuscript are not quite correct and need to be further detailed, or substituted by more general statements referring to lactate’s ability to act as a gluconeogenic precursor and an anaplerotic molecule that supports cancer cell’s biosynthetic and energetic needs for proliferation (consistently with what reported in both ref. 65 and 73 of the revised manuscript). At this regard, the occurrence of lactate oxidation by mitochondria (refs 65, 73, 75 of the revised manuscript, and refs therein) due to the mitochondrial LDH and lactate translocators such as mitochondrial MCTs, should be at least mentioned. Indeed, experimental support is actually lacking for the conversion of lactate into pyruvate in the cytosol, a thermodynamically unfavorable reaction, as well as for the notion that glycolytic flux is directed to lactate production only when oxygen is lacking and for the traditional view that glycolytic flux is directed to mitochondrial respiration solely through pyruvate uptake by the mitochondrial reticulum. Consistently, an arrow from lactate to mitochondrion should be added to Figs. 1 and 3 and figure captions should be modified accordingly.
  2. Some typos and grammatical errors are present in the new chapter 3.2, as well as throughout the manuscript. A check is required.

Author Response

Reply to Reviewer 1

Comments and Suggestions for Authors

Critical issues for the revised Manuscript:

1. Chapter 3.2: At lines 187 and 189-190 the authors state that L-lactate could be used by cancer cells to produce glucose by gluconeogenesis (“…promoting the synthesis of new glucose [65,73]”; “second, glucose is even regenerated by more oxygenated cells that reach a metabolic self-sufficiency [65,73]”). The ability of cancer cells to generate glucose from lactate, as well as from other non-carbohydrate compounds (glutamine, alanine, and glycerol), is still debated, since there are discrepancies in the effects of gluconeogenic enzymes, such as PEPCK, in different tumors (PEPCK has a tumor suppressing effect in hepatoma and renal cancer, and a protumorigenic role in lung cancer and melanoma) probably due to tissue of origin. It should be also stressed that liver and kidney are the only two organs that express all genes required for a functional gluconeogenic pathway. Other organs as colon or lung cannot produce glucose, but may engage in truncated gluconeogenesis to support their biosynthetic needs rather than to increase glucose production [Wang, Z., & Dong, C. (2018). Gluconeogenesis in Cancer: Function and Regulation of PEPCK, FBPase, and G6Pase. Trends in Cancer. doi:10.1016/j.trecan.2018.11.003]. For these reasons, the above mentioned statements in the revised version of the manuscript are not quite correct and need to be further detailed, or substituted by more general statements referring to lactate’s ability to act as a gluconeogenic precursor and an anaplerotic molecule that supports cancer cell’s biosynthetic and energetic needs for proliferation (consistently with what reported in both ref. 65 and 73 of the revised manuscript). At this regard, the occurrence of lactate oxidation by mitochondria (refs 65, 73, 75 of the revised manuscript, and refs therein) due to the mitochondrial LDH and lactate translocators such as mitochondrial MCTs, should be at least mentioned. Indeed, experimental support is actually lacking for the conversion of lactate into pyruvate in the cytosol, a thermodynamically unfavorable reaction, as well as for the notion that glycolytic flux is directed to lactate production only when oxygen is lacking and for the traditional view that glycolytic flux is directed to mitochondrial respiration solely through pyruvate uptake by the mitochondrial reticulum. Consistently, an arrow from lactate to mitochondrion should be added to Figs. 1 and 3 and figure captions should be modified accordingly.

As suggested, we substituted the sentences at lines 187- and 189-90 with more general statements indicating that lactate can act as anaplerotic molecules, supporting the energy and biosynthetic requirement s of cancer cells (line 187). Indeed, lactate is imported within mitochondria via MCTs; here the mitochondrial LDH isoform transforms lactate into pyruvate, which can act both as an energetic substrate or a building block (line 190).

As suggested we modified figure 1 and 3, and their legends accordingly.

2. Some typos and grammatical errors are present in the new chapter 3.2, as well as throughout the manuscript. A check is required.

We checked paragraph 3.2, and the whole manuscript, for typos and grammatical errors.

Reviewer 2 Report

In the current revised version of their manuscript, the authors have now addressed all my comments.

Minor points (typo):

-for Mitochondrial Pyruvate Carrier, the authors must refer to MPC instead of MCP

-for GLI2-related article, this is ref 169 (not ref 170)

-line 116: "glucose-metabolizing enzymes"

-line 117: "such as hexokinase"

-line 178: use "variable" instead of "variegate"

-line 243: "activate"

-line 362: "is downregulated"

-line 398: "metabolic symbiosis"

Author Response

Reply to Reviewer 2

Comments and Suggestions for Authors

In the current revised version of their manuscript, the authors have now addressed all my comments.

Minor points (typo):

-for Mitochondrial Pyruvate Carrier, the authors must refer to MPC instead of MCP

-for GLI2-related article, this is ref 169 (not ref 170)

-line 116: "glucose-metabolizing enzymes"

-line 117: "such as hexokinase"

-line 178: use "variable" instead of "variegate"

-line 243: "activate"

-line 362: "is downregulated"

-line 398: "metabolic symbiosis"

We thank the Reviewer for the positive comments. We corrected the typos indicated above.